# $k$-Median Clustering via Metric Embedding: Towards Better Initialization with Privacy

## Abstract

In clustering algorithms, the choice of initial centers is crucial for the quality of the learned clusters. We propose a new initialization scheme for the $k$-median problem in the general metric space (e.g., discrete space induced by graphs), based on the construction of metric embedding tree structure of the data. From the tree, we propose a novel and efficient search algorithm, for good initial centers that can be used subsequently for the local search algorithm. The so-called HST initialization method can produce initial centers achieving lower errors than those from another popular initialization method, $k$-median++, with comparable efficiency. Our HST initialization can also be easily extended to the setting of differential privacy (DP) to generate private initial centers. We show that the error of applying DP local search followed by our private HST initialization improves previous results on the approximation error, and approaches the lower bound within a small factor. Experiments demonstrate the effectiveness of our proposed methods.

## 1 Introduction

Clustering is an important problem in unsupervised learning that has been widely studied in statistics, data mining, network analysis, etc. (Punj and Stewart, 1983; Dhillon and Modha, 2001; Banerjee et al., 2005; Berkhin, 2006; Abbasi and Younis, 2007). The goal of clustering is to partition a set of data points into clusters such that items in the same cluster are expected to be similar, while items in different clusters should be different. This is concretely measured by the sum of distances (or squared distances) between each point to its nearest cluster center. One conventional notion to evaluate a clustering algorithms is: with high probability,

$$cost(C, D) \leq \gamma OPT_k(D) + \xi,$$

where $C$ is the centers output by the algorithm and $cost(C, D)$ is a cost function defined for $C$ on dataset $D$. $OPT_k(D)$ is the cost of optimal (oracle) clustering solution on $D$. When everything is clear from context, we will use $OPT$ for short. Here, $\gamma$ is called *multiplicative error* and $\xi$ is called *additive error*. Alternatively, we may also use the notion of expected cost.

Two popularly studied clustering problems are 1) the $k$-median problem, and 2) the $k$-means problem. The origin of $k$-median dates back to the 1970's (e.g., Kaufman et al. (1977)), where one tries to find the best location of facilities that minimizes the cost measured by the distance between clients and facilities. Formally, given a set of points $D$ and a distance measure, the goal is to find $k$ center points minimizing the sum of absolute distances of each sample point to its nearest center. In $k$-means, the objective is to minimize the sum of squared distances instead. Particularly, $k$-median is usually the one used for clustering on graph/network data. In general, there are two popular frameworks for clustering. One heuristic is the Lloyd's algorithm (Lloyd, 1982), which is built upon an iterative distortion minimization approach. In most cases, this method can only be applied to numerical data, typically in the (continuous) Euclidean space. Clustering in general metric spaces (discrete spaces) is also important and useful when dealing with, for example, the graph data, where Lloyd's method is no longer applicable. A more broadly applicable approach, the local search method (Kanungo et al., 2002; Arya et al., 2004), has also been widely studied. It iteratively finds the optimal swap between the center set and non-center data points to keep lowering the cost. Local search can achieve a constant approximation ratio ($\gamma = O(1)$) to the optimal solution for $k$-median (Arya et al., 2004).

**Initialization of cluster centers.** It is well-known that the performance of clustering can be highly sensitive to initialization. If clustering starts with good initial centers (i.e., with small approximation error), the algorithm may use fewer iterations to find a better solution. The $k$-median++

algorithm (Arthur and Vassilvitskii, 2007) iteratively selects $k$ data points as initial centers, favoring distant points in a probabilistic way. Intuitively, the initial centers tend to be well spread over the data points (i.e., over different clusters). The produced initial center is proved to have $O(\log k)$ multiplicative error. Follow-up works of $k$-means++ further improved its efficiency and scalability, e.g., Bahmani et al. (2012); Bachem et al. (2016); Lattanzi and Sohler (2019). In this work, we propose a new initialization framework, called HST initialization, based on metric embedding techniques. Our method is built upon a novel search algorithm on metric embedding trees, with comparable approximation error and running time as $k$-median++. Moreover, importantly, our initialization scheme can be conveniently combined with the notion of differential privacy (DP).

**Clustering with Differential Privacy.** The concept of differential privacy (Dwork, 2006; McSherry and Talwar, 2007) has been popular to rigorously define and resolve the problem of keeping useful information for model learning, while protecting privacy for each individual. Private $k$-means problem has been widely studied, e.g., Feldman et al. (2009); Nock et al. (2016); Feldman et al. (2017), mostly in the continuous Euclidean space. The paper (Balcan et al., 2017) considered identifying a good candidate set (in a private manner) of centers before applying private local search, which yields $O(\log^3 n)$ multiplicative error and $O((k^2+d)\log^5 n)$ additive error. Later on, the Euclidean $k$-means errors are further improved to $\gamma = O(1)$ and $\xi = O(k^{1.01} \cdot d^{0.51} + k^{1.5})$ by Stemmer and Kaplan (2018), with more advanced candidate set selection. Huang and Liu (2018) gave an optimal algorithm in terms of minimizing Wasserstein distance under some data separability condition.

For private $k$-median clustering, Feldman et al. (2009) considered the problem in high dimensional Euclidean space. However, it is rather difficult to extend their analysis to more general metrics in discrete spaces (e.g., on graphs). The strategy of (Balcan et al., 2017) to form a candidate center set could as well be adopted to $k$-median, which leads to $O(\log^{3/2} n)$ multiplicative error and $O((k^2 + d)\log^3 n)$ additive error in high dimensional Euclidean space. In discrete space, Gupta et al. (2010) proposed a private method for the classical local search heuristic, which applies to both $k$-medians and $k$-means. To cast privacy on each swapping step, the authors applied the exponential mechanism of (McSherry and Talwar, 2007). Their method produced an $\epsilon$-differentially private solution with cost $6OPT + O(\triangle k^2 \log^2 n/\epsilon)$, where $\triangle$ is the diameter of the point set. In this work, we will show that our HST initialization can improve DP local search for $k$-median (Gupta et al., 2010) in terms of both approximation error and efficiency.

**The main contributions** of this work include :

- We introduce the Hierarchically Well-Separated Tree (HST) to the $k$-median clustering problem for initialization. We design an efficient sampling strategy to select the initial center set from the tree, with an approximation factor $O(\log \min\{k, \triangle\})$ in the non-private setting, which is $O(\log \min\{k, d\})$ when $\triangle = O(d)$ (e.g., bounded data). This improves the $O(\log k)$ error of $k$-means/median++ in e.g., the lower dimensional Euclidean space.

- We propose a differentially private version of HST initialization under the setting of Gupta et al. (2010) in discrete metric space. The so-called DP-HST algorithm finds initial centers with $O(\log n)$ multiplicative error and $O(\epsilon^{-1}\triangle k^2 \log^2 n)$ additive error. Moreover, running DP-local search starting from this initialization gives $O(1)$ multiplicative error and $O(\epsilon^{-1}\triangle k^2 (\log \log n) \log n)$ additive error, which improves previous results towards the well-known lower bound $O(\epsilon^{-1}\triangle k \log(n/k))$ on the additive error of DP $k$-median (Gupta et al., 2010) within a small $O(k \log \log n)$ factor. This is the first clustering initialization method with differential privacy guarantee and improved error rate in general metric space.

- We conduct experiments on simulated and real-world datasets to demonstrate the effectiveness of our methods. In both non-private and private settings, our proposed HST-based approach achieves smaller cost at initialization than $k$-median++, which may also lead to improvements in the final clustering quality.

## 2 BACKGROUND AND SETUP

**Definition 2.1** (Differential Privacy (DP) (Dwork, 2006))**.** If for any two adjacent data sets $D$ and $D'$ with symmetric difference of size one, for any $O \subset Range(\mathbb{A})$, an algorithm $\mathbb{A}$ satisfies

$$Pr[\mathbb{A}(D) \in O] \le e^\epsilon Pr[\mathbb{A}(D') \in O],$$

then algorithm $\mathbb{A}$ is said to be $\epsilon$-differentially private.

Intuitively, DP requires that after removing any observation, the output of $D'$ should not be too different from that of the original dataset $D$. Smaller $\epsilon$ indicates stronger privacy, which, however, usually sacrifices utility. Thus, one central topic in DP is to balance the utility-privacy trade-off.

To achieve DP, one approach is to add noise to the algorithm output. The *Laplace mechanism* adds Laplace$(\eta(f)/\epsilon)$ noise to the output, which is known to achieve $\epsilon$-DP. The *exponential mechanism* is also a tool for many DP algorithms. Let $O$ be the set of feasible outputs. The utility function $q : D \times O \to \mathbb{R}$ is what we aim to maximize. The exponential mechanism outputs an element $o \in O$ with probability $P[\mathbb{A}(D) = o] \propto \exp(\frac{\epsilon q(D,o)}{2\eta(q)})$, where $D$ is the input dataset and $\eta(f) = \sup_{|D-D'|=1} |f(D) - f(D')|$ is the sensitivity of $f$. Both mechanisms will be used in our paper.

## 2.1 $k$-MEDIAN CLUSTERING

Following Arya et al. (2004); Gupta et al. (2010), the problem of $k$-median clustering (DP and non-DP) studied in our paper is stated as below.

**Definition 2.2** ($k$-median). Given a universe point set $U$ and a metric $\rho : U \times U \to \mathbb{R}$, the goal of $k$-median to pick $F \subseteq U$ with $|F| = k$ to minimize

$$k\text{-median:} \quad cost_k(F,U) = \sum_{v \in U} \min_{f \in F} \rho(v, f). \tag{1}$$

Let $D \subseteq U$ be a set of demand points. The goal of DP $k$-median is to minimize

$$\textbf{DP } k\text{-median:} \quad cost_k(F,D) = \sum_{v \in D} \min_{f \in F} \rho(v, f). \tag{2}$$

At the same time, the output $F$ is required to be $\epsilon$-differentially private to $D$. We may drop "$F$" and use "$cost_k(U)$" or "$cost_k(D)$" if there is no risk of ambiguity.

To better understand the motivation of the DP clustering, we provide a real-world example as follows.

**Example 2.3.** *Consider $U$ to be the universe of all users in a social network (e.g., Twitter). Each user (account) is public, but also has some private information that can only be seen by the data holder. Let $D$ be users grouped by some feature that might be set as private. Suppose a third party plans to collaborate with the most influential users in $D$ for e.g., commercial purposes, thus requesting the cluster centers of $D$. In this case, we need a strategy to safely release the centers, while protecting the individuals in $D$ from being identified (since the membership of $D$ is private).*

The local search procedure for $k$-median proposed by Arya et al. (2004) is summarized in Algorithm 1. First we randomly pick $k$ points in $U$ as the initial centers. In each iteration, we search over all $x \in F$ and $y \in U$, and do the swap $F \leftarrow F - \{x\} + \{y\}$ such that $F - \{x\} + \{y\}$ improves the cost of $F$ the most (if more than factor $(1 - \alpha/k)$ where $\alpha > 0$ is a hyper-parameter). We repeat the procedure until no such swap exists. Arya et al. (2004) showed that the output centers $F$ achieves 5 approximation error to the optimal solution, i.e., $cost(F) \leq 5OPT$.

---

**Algorithm 1:** Local search for $k$-median clustering (Arya et al., 2004)

**Input:** Data points $U$, parameter $k$, constant $\alpha$
**Initialization:** Randomly select $k$ points from $U$ as initial center set $F$
**while** $\exists x \in F, y \in U$ *s.t.* $cost(F - \{x\} + \{y\}) \leq (1 - \alpha/k)cost(F)$ **do**
  Select $(x,y) \in F_i \times (D \setminus F_i)$ with $\arg\min_{x,y}\{cost(F - \{x\} + \{y\})\}$
  Swap operation: $F \leftarrow F - \{x\} + \{y\}$
**Output:** Center set $F$

---

## 2.2 $k$-MEDIAN++ INITIALIZATION

Although local search is able to find a solution with constant error, it takes $O(n^2)$ per iteration (Resende and Werneck, 2007) in expected $O(k \log n)$ steps (in total $O(kn^2 \log n)$) when started from random center set, which would be slow for large datasets. Indeed, we do not need such complicated algorithm to reduce the cost at the beginning, i.e., when the cost is large. To accelerate the process,

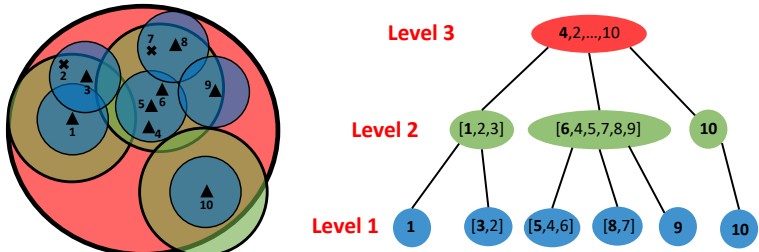

Figure 1: An illustrative example of a 3-level padded decomposition and corresponding 2-HST. **Left:** The thickness of the ball represents the level. The color corresponds to the levels in the HST in the right panel. "$\triangle$"'s are the center nodes of partitions (balls), and "$\times$"'s are non-center data points. **Right:** The resulting 2-HST generated from the padded decomposition.

efficient initialization methods find a "roughly" good center set as the starting point for local search. In the paper, we compare our new initialization scheme mainly with a popular (and perhaps most well-known) initialization method, the $k$-median++ (Arthur and Vassilvitskii, 2007) (see Algorithm 6 in the Appendix). Arthur and Vassilvitskii (2007) showed that the output centers $C$ by $k$-median++ achieves $O(\log k)$ approximation error with time complexity $O(nk)$. Starting from the initialization, we only need to run $O(k \log \log k)$ steps of the computationally heavy local search to reach a constant error solution. Thus, initialization may greatly improve the clustering efficiency.

# 3 INITIALIZATION VIA HIERARCHICALLY WELL-SEPARATED TREE (HST)

In this section, we propose our novel initialization scheme for $k$-median clustering, and provide our analysis in the non-private case solving (1). The idea is based on the metric embedding theory. We will start with an introduction to the main tool used in our approach.

## 3.1 HIERARCHICALLY WELL-SEPARATED TREE (HST)

In this paper, for an $L$-level tree, we will count levels in descending order down the tree. We use $h_v$ to denote the level of $v$, and $n_i$ be the number of nodes at level $i$. The Hierarchically Well-Separated Tree (HST) is based on the padded decompositions of a general metric space in a hierarchical manner (Fakcharoenphol et al., 2004). Let $(U, \rho)$ be a metric space with $|U| = n$, and we will refer to this metric space without specific clarification. A $\beta$–padded decomposition of $U$ is a probabilistic distribution of partitions of $U$ such that the diameter of each cluster $U_i \in U$ is at most $\beta$, i.e., $\rho(u, v) \leq \beta, \forall u, v \in U_i, i = 1, ..., k$. The formal definition of HST is given as below.

**Definition 3.1.** Assume $\min_{u,v \in U} \rho(u, v) = 1$ and denote $\triangle = \max_{u,v \in U} \rho(u, v)$. An $\alpha$-Hierarchically Well-Separated Tree ($\alpha$-HST) with depth $L$ is an edge-weighted rooted tree $T$, such that an edge between any pair of two nodes of level $i - 1$ and level $i$ has length at most $\triangle/\alpha^{L-i}$.

In this paper, we consider $\alpha = 2$-HST for simplicity, as $\alpha$ only affects the constants in our theoretical analysis. Figure 1 is an example $L = 3$-level 2-HST (right panel), along with its underlying padded decompositions (left panel). A 2-HST can be built as follows: we first find a padded decomposition $P_L = \{P_{L,1}, ..., P_{L,n_L}\}$ of $U$ with parameter $\beta = \triangle/2$. The center of each partition in $P_{L,j}$ serves as a root node in level $L$. Then, we re-do a padded decomposition for each partition $P_{L,j}$, to find sub-partitions with diameter $\beta = \triangle/4$, and set the corresponding centers as the nodes in level $L - 1$, and so on. Each partition at level $i$ is obtained with $\beta = \triangle/2^{L-i}$. This process proceeds until a node has a single point (leaf), or a pre-specified tree depth is reached. More details can be found in Algorithm 7 in the Appendix A. Blelloch et al. (2017) proposed an efficient HST construction in $O(m \log n)$ time, where $n$ and $m$ are the number of nodes and edges in a graph, respectively.

The first step of our method is to embed the data points into a HST (see Algorithm 2). Next, we will describe our new strategy to search for the initial centers on the tree (w.r.t. the tree metric). Before moving on, it is worth mentioning that, there are polynomial time algorithms for computing an *exact* $k$-median solution in the tree metric (Tamir (1996); Shah (2003)). However, the dynamic programming algorithms have high complexity (e.g., $O(kn^2)$), making them unsuitable for the purpose of fast initialization. Moreover, it is unknown how to apply them effectively to the private

---

**Algorithm 2:** NDP-HST initialization

---

**Input:** $U, \triangle, k$
**Initialization:** $L = \log \triangle, C_0 = \emptyset, C_1 = \emptyset$
Call Algorithm 7 to build a level-$L$ 2-HST $T$ using $U$
**for** *each node $v$ in $T$* **do**
    $N_v \leftarrow |U \cap T(v)|$
    $score(v) \leftarrow N_v \cdot 2^{h_v}$
**while** $|C_1| < k$ **do**
    Add top $(k - |C_1|)$ nodes with highest score to $C_1$
    **for** *each $v \in C_1$* **do**
        $C_1 = C_1 \setminus \{v\}$, if $\exists\, v' \in C_1$ such that $v'$ is a descendant of $v$
$C_0 = $ FIND-LEAF$(T, C_1)$
**Output:** Initial center set $C_0 \subseteq U$

---

---

**Algorithm 3:** FIND-LEAF $(T, C_1)$

---

**Input:** $T, C_1$
**Initialization:** $C_0 = \emptyset$
**for** *each node $v$ in $C_1$* **do**
    **while** *$v$ is not a leaf node* **do**
        $v \leftarrow \arg_w \max\{N_w, w \in ch(v)\}$, where $ch(v)$ denotes the children nodes of $v$
    Add $v$ to $C_0$
**Output:** Initial center set $C_0 \subseteq U$

---

case. As will be shown, our new algorithm 1) is very efficient, 2) gives $O(1)$ approximation error in the tree metric, and 3) can be effectively extended to DP easily.

## 3.2 HST INITIALIZATION ALGORITHM

Let $L = \log \Delta$ and suppose $T$ is a level-$L$ 2-HST in $(U, \rho)$, where we assume $L$ is an integer. For a node $v$ at level $i$, we use $T(v)$ to denote the subtree rooted at $v$. Let $N_v = |T(v)|$ be the number of data points in $T(v)$. The search strategy for the initial centers, NDP-HST initialization ("NDP" stands for "Non-Differentially Private"), is presented in Algorithm 2 with two phases.

**Subtree search.** The first step is to identify the subtrees that contain the $k$ centers. To begin with, $k$ initial centers $C_1$ are picked from $T$ who have the largest $score(v) = N(v) \cdot 2^{h_v}$. This is intuitive, since to get a good clustering, we typically want the ball surrounding each center to include more data points. Next, we do a screening over $C_1$: if there is any ancestor-descendant pair of nodes, we remove the ancestor from $C_1$. If the current size of $C_1$ is smaller than $k$, we repeat the process until $k$ centers are chosen (we do not re-select nodes in $C_1$ and their ancestors). This way, $C_1$ contains $k$ root nodes of $k$ disjoint subtrees.

**Leaf search.** After we find $C_1$ the set of $k$ subtrees, the next step is to find the center in each subtree using Algorithm 3 ("FIND-LEAF"). We employ a greedy search strategy, by finding the child node with largest score level by level, until a leaf is found. This approach is intuitive since the diameter of the partition ball exponentially decays with the level. Therefore, we are in a sense focusing more and more on the region with higher density (i.e., with more data points).

The complexity of our search algorithm is given as follows.

**Proposition 3.2** (Complexity). *Algorithm 2 takes $O(dn \log n)$ time in the Euclidean space.*

**Remark 3.3.** The complexity of HST initialization is in general comparable to $O(dnk)$ of $k$-median++. Our algorithm would be faster if $k > \log n$, i.e., the number of centers is large.

## 3.3 APPROXIMATION ERROR OF HST INITIALIZATION

Firstly, we show that the initial center set produced by NDP-HST is already a good approximation to the optimal $k$-median solution. Let $\rho^T(x, y) = d_T(x, y)$ denote the "2-HST metric" between $x$ and

$y$ in the 2-HST $T$, where $d_T(x, y)$ is the tree distance between nodes $x$ and $y$ in $T$. By Definition 3.1 and since $\triangle = 2^L$, in the analysis we assume equivalently that the edge weight of the $i$-th level $2^{i-1}$. The crucial step of our analysis is to examine the approximation error in terms of the 2-HST metric, after which the error can be adapted to the general metrics by the following Lemma (Bartal, 1996).

**Lemma 3.4.** *In a metric space $(U, \rho)$ with $|U| = n$ and diameter $\triangle$, it holds that $E[\rho^T(x, y)] = O(\min\{\log n, \log \triangle\})\rho(x, y)$. In the Euclidean space $\mathbb{R}^d$, $E[\rho^T(x, y)] = O(d)\rho(x, y)$.*

Recall $C_0, C_1$ from Algorithm 2. We define

$$cost_k^T(U) = \sum_{y \in U} \min_{x \in C_0} \rho^T(x, y), \tag{3}$$

$$cost_k^{T'}(U, C_1) = \min_{\substack{|F \cap T(v)| = 1, \\ \forall v \in C_1}} \sum_{y \in U} \min_{x \in F} \rho^T(x, y), \tag{4}$$

$$OPT_k^T(U) = \min_{F \subset U, |F| = k} \sum_{y \in U} \min_{x \in F} \rho^T(x, y) \equiv \min_{C_1'} cost_k^{T'}(U, C_1'). \tag{5}$$

For simplicity, we will use $cost_k^{T'}(U)$ to denote $cost_k^{T'}(U, C_1)$. Here, $OPT_k^T$ (5) is the cost of the global optimal solution with 2-HST metric. The last equivalence in (5) holds because the optimal centers set can always located in $k$ disjoint subtrees, as each leaf only contain one point. (3) is the $k$-median cost with 2-HST metric of the output $C_0$ of Algorithm 2. (4) is the oracle cost after the subtrees are chosen. That is, it represents the optimal cost to pick one center from each subtree in $C_1$. Firstly, we bound the approximation error of subtree search and leaf search, respectively.

**Lemma 3.5** (Subtree search). $cost_k^{T'}(U) \le 5OPT_k^T(U)$.

**Lemma 3.6** (Leaf search). $cost_k^T(U) \le 2cost_k^{T'}(U)$.

Combining Lemma 3.5 and Lemma 3.6, we obtain

**Theorem 3.7** (2-HST error). *Running Algorithm 2, we have $cost_k^T(U) \le 10OPT_k^T(U)$.*

Thus, HST-initialization produces an $O(1)$ approximation to $OPT$ in the 2-HST metric. Define $cost_k(U)$ as (1) for our HST centers, and the optimal cost w.r.t. $\rho$ as

$$OPT_k(U) = \min_{|F| = k} \sum_{y \in U} \min_{x \in F} \rho(x, y). \tag{6}$$

We have the following result based on Lemma 3.4.

**Theorem 3.8.** *In general metric space, the expected $k$-median cost of Algorithm 2 is $E[cost_k(U)] = O(\min\{\log n, \log \triangle\})OPT_k(U)$.*

**Remark 3.9.** In the Euclidean space, Makarychev et al. (2019) proved $O(\log k)$ random projections suffice for $k$-median to achieve $O(1)$ error. Thus, if $\triangle = O(d)$ (e.g., bounded data), by Lemma 3.4, HST initialization is able to achieve $O(\log(\min\{d, k\}))$ error, which is better than $O(\log k)$ of $k$-median++ when $d$ is small.

**NDP-HST Local Search.** We are interested in the approximation quality of standard local search (Algorithm 1), when initialized by our NDP-HST.

**Theorem 3.10.** *NDP-HST local search achieves $O(1)$ approximation error in expected $O(k \log \log \min\{n, \triangle\})$ number of iterations for input in general metric space.*

Before ending this section, we remark that the initial centers found by NDP-HST can be used for $k$-means clustering analogously. For general metrics, $E[cost_{km}(U)] = O(\min\{\log n, \log \triangle\})^2 OPT_{km}(U)$ where $cost_{km}(U)$ is the optimal $k$-means cost. See Appendix D for the detailed (and similar) analysis.

## 4 HST INITIALIZATION WITH DIFFERENTIAL PRIVACY

In this section, we consider initialization method with differential privacy (DP). Recall (2) that $U$ is the universe of data points, and $D \subset U$ is a demand set that needs to be clustered with privacy.

---

**Algorithm 4:** DP-HST initialization

---

**Input:** $U, D, \triangle, k, \epsilon$
Build a level-$L$ 2-HST $T$ based on input $U$
**for** *each node $v$ in $T$* **do**
  $N_v \leftarrow |D \cap T(v)|$
  $\hat{N}_v \leftarrow N_v + Lap(2^{(L-h_v)}/\epsilon)$
  $score(v) \leftarrow \hat{N}(v) \cdot 2^{h_v}$
Based on $\hat{N}_v$, apply the same strategy as Algorithm 2: find $C_1$; $C_0 = $ FIND-LEAF($C_1$)
**Output:** Private initial center set $C_0 \subseteq U$

---

**Algorithm 5:** DP-HST local search

---

**Input:** $U$, demand points $D \subseteq U$, parameter $k, \epsilon, T$
**Initialization:** $F_1$ the private initial centers generated by Algorithm 4 with privacy $\epsilon/2$
Set parameter $\epsilon' = \frac{\epsilon}{4\triangle(T+1)}$
**for** $i = 1$ *to $T$* **do**
  Select $(x, y) \in F_i \times (V \setminus F_i)$ with prob. proportional to $\exp(-\epsilon' \times (cost(F_i - \{x\} + \{y\})))$
  Let $F_{i+1} \leftarrow F_i - \{x\} + \{y\}$
Select $j$ from $\{1, 2, ..., T+1\}$ with probability proportional to $\exp(-\epsilon' \times cost(F_j))$
**Output:** $F = F_j$ the private center set

---

Since $U$ is public, simply running initialization algorithms on $U$ would preserve the privacy of $D$. However, 1) this might be too expensive; 2) in many cases one would probably want to incorporate some information about $D$ in the initialization, since $D$ could be a very imbalanced subset of $U$. For example, $D$ may only contain data points from one cluster, out of tens of clusters in $U$. In this case, initialization on $U$ is likely to pick initial centers in multiple clusters, which would not be helpful for clustering on $D$. Next, we show how our proposed HST initialization can be easily combined with differential privacy that at the same time contains information about the demand set $D$, leading to improved approximation error (Theorem 4.3). Again, suppose $T$ is an $L = \log \triangle$-level 2-HST of universe $U$ in a general metric space. Denote $N_v = |T(v) \cap D|$ for a node point $v$. Our private HST initialization (DP-HST) is similar to the non-private Algorithm 2. To gain privacy, we perturb $N_v$ by adding i.i.d. Laplace noise:

$$\hat{N}_v = N_v + Lap(2^{(L-h_v)}/\epsilon),$$

where $Lap(2^{(L-h_v)}/\epsilon)$ is a Laplace random number with rate $2^{(L-h_v)}/\epsilon$. We will use the perturbed $\hat{N}_v$ for node sampling instead of the true value $N_v$, as described in Algorithm 4. The DP guarantee of this initialization scheme is straightforward by the composition theory (Dwork, 2006).

**Theorem 4.1.** *Algorithm 4 is $\epsilon$-differentially private.*

*Proof.* For each level $i$, the subtrees $T(v, i)$ are disjoint to each other. The privacy used in $i$-th level is $\epsilon/2^{(L-i)}$, and the total privacy is $\sum_i \epsilon/2^{(L-i)} < \epsilon$. $\square$

We now consider the approximation error. As the structure of the analysis is similar to the non-DP case, we present the main result here and defer the detailed proofs to Appendix C.

**Theorem 4.2.** *Algorithm 4 finds initial centers such that*

$$E[cost_k(D)] = O(\log n)(OPT_k(D) + k\epsilon^{-1}\triangle \log n).$$

**DP-HST Local Search.** Similarly, we can use private HST initialization to improve the performance of private $k$-median local search, which is presented in Algorithm 5. After initialization, the DP local search procedure follows Gupta et al. (2010) using the exponential mechanism.

**Theorem 4.3.** *Algorithm 5 achieves $\epsilon$-differential privacy. With probability $(1 - \frac{1}{poly(n)})$, the output centers admit*

$$cost_k(D) \leq 6OPT_k(D) + O(\epsilon^{-1}k^2\triangle(\log \log n) \log n)$$

*in $T = O(k \log \log n)$ iterations.*

The DP local search with random initialization (Gupta et al., 2010) has 6 multiplicative error and $O(\epsilon^{-1} \triangle k^2 \log^2 n)$ additive error. Our result improves the $\log n$ term to $\log \log n$ in the additive error. Meanwhile, the number of iterations needed is improved from $T = O(k \log n)$ to $O(k \log \log n)$ (see Appendix B for an empirical justification). Notably, it has been shown in Gupta et al. (2010) that for $k$-median problem, the lower bounds on the multiplicative and additive error of any $\epsilon$-DP algorithm are $O(1)$ and $O(\epsilon^{-1} \triangle k \log(n/k))$, respectively. Our result matches the lower bound on the multiplicative error, and the additive error is only worse than the bound by a factor of $O(k \log \log n)$ which would be small in many cases. To our knowledge, Theorem 4.3 is the first result in literature to improve the error of DP local search in general metric space.

## 5 EXPERIMENTS

### 5.1 DATASETS AND ALGORITHMS

**Discrete Euclidean space.** Following previous work ., we test $k$-median clustering on the MNIST hand-written digit dataset (LeCun et al., 1998) with 10 natural clusters (digit 0 to 9). We set $U$ as 10000 randomly chosen data points. We choose the demand set $D$ using two strategies: 1) "balance", where we randomly choose 500 samples from $U$; 2) "imbalance", where $D$ contains 500 random samples from $U$ only from digit "0" and "8" (two clusters). We note that, the imbalanced $D$ is a very practical setting in real-world scenarios, where data are typically not uniformly distributed. On this dataset, we test clustering with both $l_1$ and $l_2$ distance as the underlying metric.

**Metric space induced by graph.** Random graphs have been widely considered in testing $k$-median methods (Balcan et al., 2013; Todo et al., 2019). The construction of graphs follows a similar approach as the synthetic *pmedinfo* graphs provided by the popular OR-Library (Beasley, 1990). The metric $\rho$ for this experiment is the shortest (weighted) path distance. To generate a size $n$ graph, we first randomly split the nodes into 10 clusters. Within each cluster, each pair of nodes is connected with probability 0.2 and weight drawn from standard uniform distribution. For each pair of clusters, we randomly connect some nodes from each cluster, with weights following uniform $[0.5, r]$. A larger $r$ makes the graph more separable, i.e., clusters are farther from each other (see Appendix B for example graphs). We present two cases: $r = 1$ and $r = 100$. For this task, $U$ has 3000 nodes, and the private set $D$ (500 nodes) is chosen using similar "balanced" and "imbalanced" scheme as described above. In the imbalanced case, we choose $D$ randomly from only two clusters.

**Algorithms.** We compare the following clustering algorithms in both non-DP and DP setting: (1) **NDP-rand:** Local search with random initialization; (2) **NDP-kmedian++:** Local search with $k$-median++ initialization (Algorithm 6); (3) **NDP-HST:** Local search with NDP-HST initialization (Algorithm 2), as described in Section 3; (4) **DP-rand:** Standard DP local search algorithm (Gupta et al., 2010), which is Algorithm 5 with initial centers randomly chosen from $U$; (5) **DP-kmedian++:** DP local search with $k$-median++ initialization run on $U$; (6) **DP-HST:** DP local search with HST-initialization (Algorithm 5). For non-DP tasks, we set $L = 6$. For DP clustering, we use $L = 8$.

For non-DP methods, we set $\alpha = 10^{-3}$ in Algorithm 1 and the maximum number of iterations as 20. To examine the quality of initialization as well as the final centers, We report both the cost at initialization and the cost of the final output. For DP methods, we run the algorithms for $T = 20$ steps and report the results with $\epsilon = 1$. We test $k \in \{2, 5, 10, 15, 20\}$. The average cost over $T$ iterations is reported for more robustness. All results are averaged over 10 independent repetitions.

### 5.2 RESULTS

The results on MNIST dataset are given in Figure 2. The comparisons are similar for both $l_1$ and $l_2$:

- From the left column, the initial centers found by HST has lower cost than $k$-median++ and random initialization, for both non-DP and DP setting, and for both balanced and imbalanced demand set $D$. This confirms that the proposed HST initialization is more powerful than $k$-median++ in finding good initial centers.

- From the right column, we also observe lower final cost of HST followed by local search in DP clustering. In the non-DP case, the final cost curves overlap, which means that despite HST offers better initial centers, local search can always find a good solution eventually.

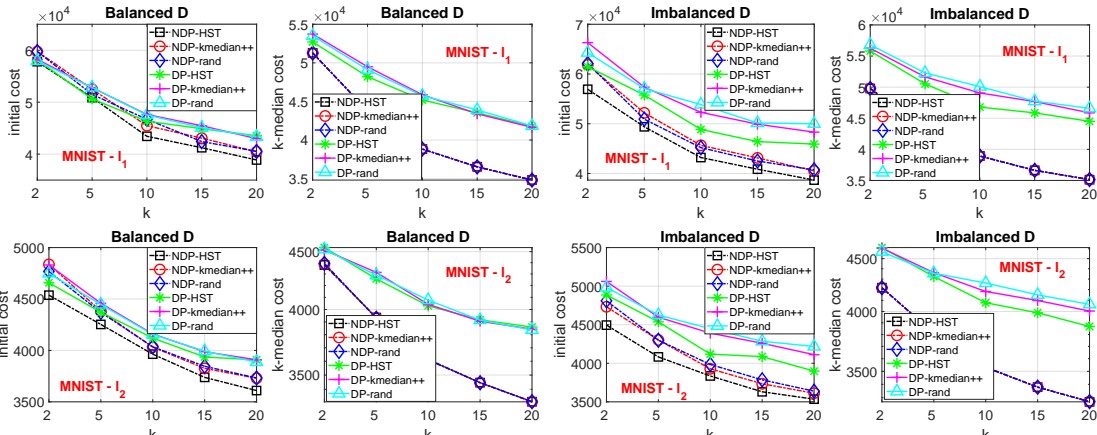

Figure 2: $k$-median cost on MNIST dataset. **1st column:** initial cost. **2nd column:** final output cost.

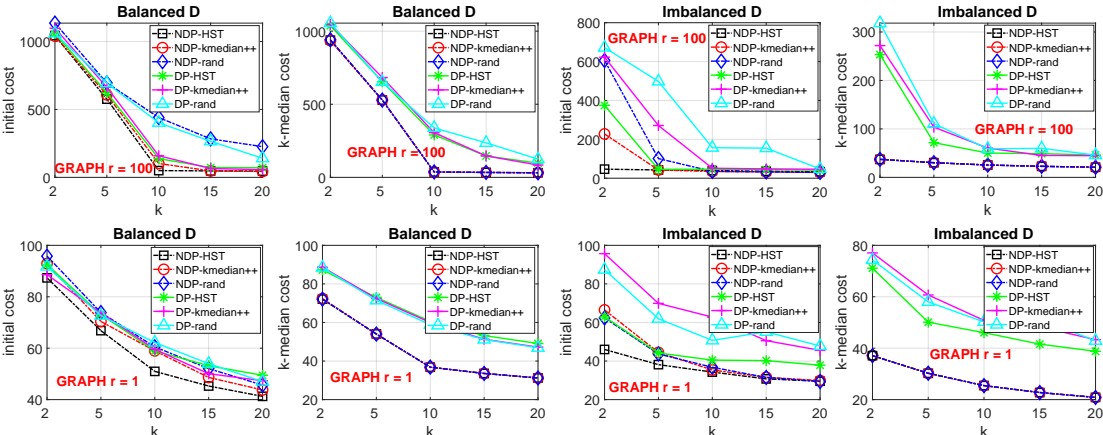

Figure 3: $k$-median cost on graph dataset. **1st column:** initial cost. **2nd column:** final output cost.

- The advantage of DP-HST, in terms of both the initial and the final cost, is more significant when $D$ is an imbalanced subset of $U$. As mentioned before, this is because our DP-HST initialization approach also privately incorporates the information of $D$.

The results on graphs are reported in Figure 3, which give similar conclusions. In all cases, our proposed HST scheme finds better initial centers with smaller cost than $k$-median++. Moreover, HST again considerably outperforms $k$-median++ in the private and imbalanced $D$ setting, for both $r = 100$ (highly separable) and $r = 1$ (less separable). The advantages of HST over $k$-median++ are especially significant in the harder tasks when $r = 1$, i.e., the clusters are nearly mixed up.

## 6    CONCLUSION

In this paper, we propose a new initialization framework for the $k$-median problem in general metric space. Our approach is called HST initialization, which leverages tools from metric embedding theory. Our novel tree search approach has comparable efficiency and approximation error to the popular $k$-median++ initialization. Moreover, we propose differentially private (DP) HST initialization algorithm, which adapts to the private demand point set, leading to better clustering performance. When combined with subsequent DP local search heuristic, our algorithm is able to improve the additive error of DP local search, which is close to the theoretical lower bound within a small factor. Experiments with Euclidean metrics and graph metrics verify the effectiveness of our methods, which improve the cost of both the initial centers and the final $k$-median output.

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

# Supplemental Materials

## A  POSTPONED ALGORITHM

### A.1  $k$-MEDIAN++

In the paper, we compared our HST initialization mainly with another (perhaps most well-known) initialization algorithm for clustering, the $k$-median++ (Arthur and Vassilvitskii, 2007). For reference, we present the concrete procedures in Algorithm 6. Here, the function $D(u, C)$ is the shortest distance from a data point $u$ to the closest (center) point in set $C$. Arthur and Vassilvitskii (2007) showed that the output centers $C$ by $k$-median++ achieves $O(\log k)$ approximation error, in $O(dnk)$ time.

---

**Algorithm 6:** $k$-median++ initialization (Arthur and Vassilvitskii, 2007)

---

**Input:** Data points $U$, number of centers $k$
Randomly pick a point $c_1 \in U$ and set $F = \{c_1\}$
**for** $i = 2$ *to* $k$ **do**
  Select $c_i = u \in U$ with probability $\frac{\rho(u,F)}{\sum_{u' \in U} \rho(u',F)}$
  $F = F \cup \{c_i\}$
**Output:** $k$-median++ initial center set $F$

---

### A.2  HST CONSTRUCTION

As presented in Algorithm 7, the construction starts by applying a permutation $\pi$ on $U$, such that in following steps the points are picked in a random sequence. We first find a padded decomposition $P_L = \{P_{L,1}, ..., P_{L,n_L}\}$ of $U$ with parameter $\beta = \triangle/2$. The center of each partition in $P_{L,j}$ serves as a root node in level $L$. Then, we re-do a padded decomposition for each partition $P_{L,j}$, to find sub-partitions with diameter $\beta = \triangle/4$, and set the corresponding centers as the nodes in level $L - 1$, and so on. Each partition at level $i$ is obtained with $\beta = \triangle/2^{L-i}$. This process proceeds until a node has a single point, or a pre-specified tree depth is reached. In Figure 1, we provide an example of $L = 3$-level 2-HST (left panel), along with its underlying padded decompositions (right panel).

---

**Algorithm 7:** Build 2-HST$(U, L)$

---

**Input:** Data points $U$ with diameter $\triangle$, $L$
Randomly pick a point in $U$ as the root node of $T$
Let $r = \triangle/2$
Apply a permutation $\pi$ on $U$  `// so points will be chosen in a random sequence`
**for** *each* $v \in U$ **do**
  Set $C_v = [v]$
  **for** *each* $u \in U$ **do**
    Add $u \in U$ to $C_v$ if $d(v,u) \leq r$ and $u \notin \bigcup_{v' \neq v} C_{v'}$
Set the non-empty clusters $C_v$ as the children nodes of $T$
**for** *each non-empty cluster* $C_v$ **do**
  Run 2-HST$(C_v, L - 1)$ to extend the tree $T$; stop until $L$ levels or reaching a leaf node
**Output:** 2-HST $T$

---

# B MORE EXPERIMENTS

## B.1 EXAMPLES OF GRAPH DATA

In Figure 4, we plot two example graphs (subgraphs of 50 nodes) with $r = 100$ and $r = 1$. When $r = 100$, the graph is highly separable (i.e., clusters are far from each other). When $r = 1$, the clusters are harder to be distinguished from each other.

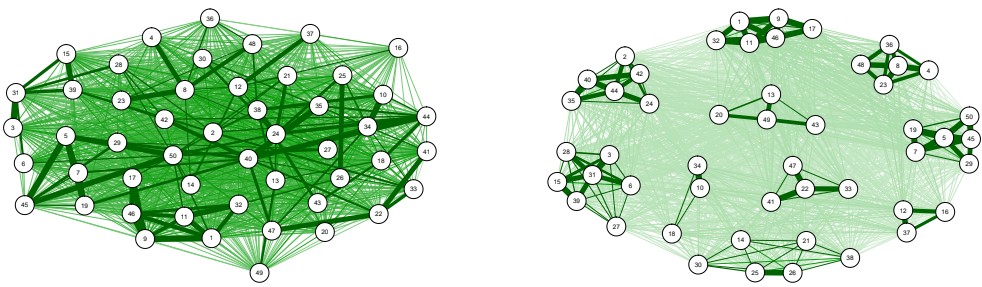

Figure 4: Example of synthetic graphs: subgraph of 50 nodes. **Left:** $r = 1$. **Right:** $r = 100$. Darker and thicker edged have smaller distance. When $r = 100$, the graph is more separable.

## B.2 RUNNING TIME COMPARISON WITH $k$-MEDIAN++

In Proposition 3.2, we show that our HST initialization algorithm admits $O(dn \log n)$ complexity when considering the Euclidean space. With a smart implementation of Algorithm 6 where each data point tracks its distance to the current closest candidate center in $C$, $k$-median++ has $O(dnk)$ running time. Therefore, the running time of our algorithm is in general comparable to $k$-median++. Our method would run faster if $k = \Omega(\log n)$. In Figure 5, we plot the empirical running time of HST initialization against $k$-median++, on MNIST dataset with $l_2$ distance (similar comparison holds for $l_1$). From the left subfigure, we see that $k$-median++ becomes slower with increasing $k$, and our method is more efficient when $k > 20$. In the right panel, we observe that the running time of both methods increases with larger sample size $n$. Our HST algorithm has a slightly faster increasing rate, which is predicted by the complexity comparison ($n \log n$ v.s. $n$). However, this difference in $\log n$ factor would not be too significant unless the sample size is extremely large. Overall, our numerical results suggest that in general, the proposed HST initialization would have similar efficiency as $k$-median++ in common practical scenarios.

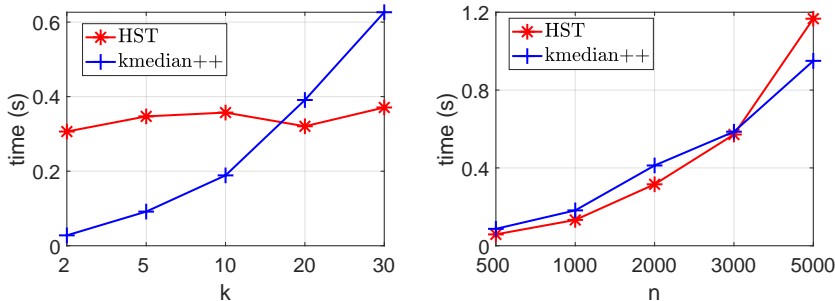

Figure 5: Empirical time comparision of HST initialization v.s. $k$-median++, on MNIST dataset with $l_2$ distance. **Left:** The running time against $k$, on a subset of $n = 2000$ data points. **Right:** The running time against $n$, with $k = 20$ centers.

### B.3 IMPROVED ITERATION COST OF DP-HST

In Theorem 4.3, we show that under differential privacy constraints, the proposed DP-HST (Algorithm 5) improves both the approximation error and the number of iterations required to find a good solution of classical DP local search (Gupta et al., 2010). In this section, we provide some numerical results to justify the theory.

First, we need to properly measure the iteration cost of DP local search. This is because, unlike the non-private clustering, the $k$-median cost after each iteration in DP local search is not decreasing monotonically, due to the probabilistic exponential mechanism. To this end, for the cost sequence with length $T = 20$, we compute its moving average sequence with window size 5. Attaining the minimal value of the moving average indicates that the algorithm has found a "local optimum", i.e., it has reached a "neighborhood" of solutions with small clustering cost. Thus, we use the number of iterations to reach such local optimum as the measure of iteration cost. The results are provided in Figure 6. We see that on all the tasks (MNIST with $l_1$ and $l_2$ distance, and graph dataset with $r = 1$ and $r = 100$), DP-HST has significantly smaller iterations cost. In Figure 7, we further report the $k$-median cost of the best solution in $T$ iterations found by each DP algorithm. We see that DP-HST again provide the smallest cost. This additional set of experiments again validates the claims of Theorem 4.3, that DP-HST is able to found better initial centers in fewer iterations.

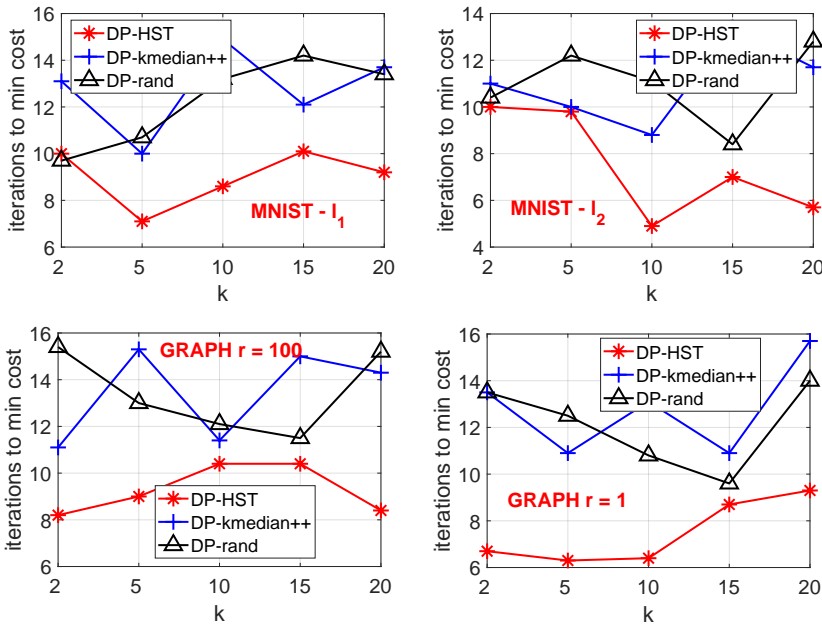

Figure 6: Iteration cost to reach a locally optimal solution, on MNIST and graph datasets with different $k$. The demand set is an imbalanced subset of the universe.

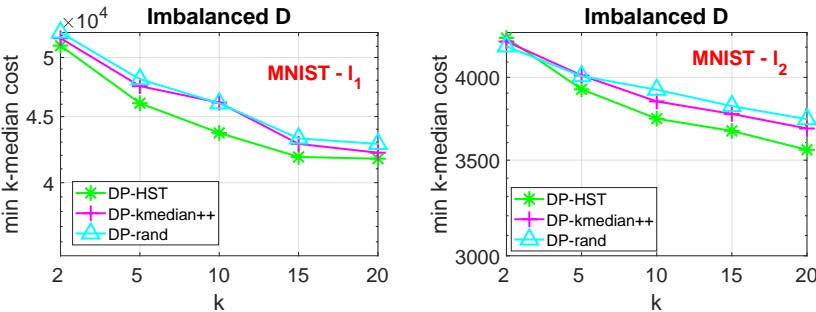

Figure 7: The $k$-median cost of the best solution found by each differentially private algorithm. The demand set is an imbalanced subset of the universe. Same comparison holds on graph data.

# C   PROOFS

The following composition result of differential privacy will be used in our proof.

**Theorem C.1** (Composition Theorem (Dwork, 2006)). *If Algorithms $\mathbb{A}_1, \mathbb{A}_2, ..., \mathbb{A}_m$ are $\epsilon_1, \epsilon_2, ..., \epsilon_m$ differentially private respectively, then the union $(\mathbb{A}_1(D), \mathbb{A}_2(D), ..., \mathbb{A}_m(D))$ is $\sum_{i=1}^m \epsilon_i$-DP.*

## C.1   PROOF OF LEMMA 3.5

*Proof.* Consider the intermediate output of Algorithm 2, $C_1 = \{v_1, v_2, ..., v_k\}$, which is the set of roots of the minimal subtrees each containing exactly one output center $C_0$. Suppose one of the optimal "root set" that minimizes (4) is $C_1^* = \{v_1', v_2', ..., v_k'\}$. If $C_1 = C_1^*$, the proof is done. Thus, we prove the case for $C_1 \neq C_1^*$. Note that $T(v), v \in C_1$ are disjoint subtrees. We have the following reasoning.

- Case 1: for some $i, j'$, $v_i$ is a descendant node of $v_j'$. Since the optimal center point $f^*$ is a leaf node by the definition of (4), we know that there must exist one child node of $v_j'$ that expands a subtree which contains $f^*$. Therefore, we can always replace $v_j'$ by one of its child nodes. Hence, we can assume that $v_i$ is not a descendant of $v_j'$.

  Note that, we have $score(v_j') \leq score(v_i)$ if $v_j' \notin C_1^* \cap C_1$. Algorithm 2 sorts all the nodes based on cost value, and it would have more priority to pick $v_j'$ than $v_i$ if $score(v_j') > score(v_i)$ and $v_i$ is not a child node of $v_j'$.

- Case 2: for some $i, j'$, $v_j'$ is a descendant of $v_i$. In this case, optimal center point $f^*$, which is a leaf of $T(v_i)$, must also be a leaf node of $T(v_j')$. We can simply replace $C_1$ with the swap $C_1 \setminus \{v_i\} + \{v_j'\}$ which does not change $cost_k^{T'}(U)$. Hence, we can assume that $v_j'$ is not a descendant of $v_i$.

- Case 3: Otherwise. By the construction of $C_1$, we know that $score(v_j') \leq \min\{score(v_i), i = 1, ..., k\}$ when $v_j' \in C_1^* \setminus C_1$. Consider the swap between $C_1$ and $C_1^*$. By the definition of tree distance, we have $OPT_k^T(U) \geq \sum_{v_i \in C_1 \setminus C_1^*} N_{v_i} 2^{h_{v_i}}$, since $\{T(v_i), v_i \in C_1 \setminus C_1^*\}$ does not contain any center of the optimal solution determined by $C_1^*$ (which is also the optimal "root set" for $OPT_k^T(U)$).

Thus, we only need to consider Case 3. Let us consider the optimal clustering with center set be $C^* = \{c_1^*, c_2^*, ..., c_k^*\}$ (each center $c_j^*$ is a leaf of subtree whose root be $c_j'$), and $S_j'$ be the leaves assigned to $c_j^*$. Let $S_j$ denote the set of leaves in $S_j'$ whose distance to $c_j^*$ is strictly smaller than its distance to any centers in $C_1$. Let $P_j$ denote the union of paths between leaves of $S_j$ to its closest center in $C_1$. Let $v_j''$ be the nodes in $P_j$ with highest level satisfying $T(v_j'') \cap C_1 = \emptyset$. The score of $v_j''$ is $2^{h_{v_j''}} N(v_j'')$. That means the swap with a center $v_j'$ into $C_1$ can only reduce $4 \cdot 2^{h_{v_j''}} N(v_j'')$ to $cost_k^{T'}(U)$ (the tree distance between any leaf in $S_j$ and its closest center in $C_1$ is at most $4 \cdot 2^{h_{v_j''}}$). We just use $v_j'$ to represent $v_j''$ for later part of this proof for simplicity. By our reasoning, summing all the swaps over $C_1^* \setminus C_1$ gives

$$cost_k^{T'}(U) - OPT_k^T(U) \leq 4 \sum_{v_j' \in C_1^* \setminus C_1} N_{v_j'} 2^{h_{v_j'}},$$

$$OPT_k^T(U) \geq \sum_{v_i \in C_1 \setminus C_1^*} N_{v_i} 2^{h_{v_i}}.$$

Also, based on our discussion on Case 1, it holds that

$$N_{v_j'} 2^{h_{v_j'}} - N_{v_i} 2^{h_{v_i}} \leq 0.$$

Summing them together, we have $cost_k^{T'}(U) \leq 5 OPT_k^T(U)$. ◻

## C.2 PROOF OF LEMMA 3.6

*Proof.* Since the subtrees in $C_1$ are disjoint, it suffices to consider one subtree with root $v$. With a little abuse of notation, let $cost_1^{T'}(v, U)$ denote the optimal $k$-median cost within the point set $T(v)$ with one center in 2-HST:

$$cost_1^{T'}(v, U) = \min_{x \in T(v)} \sum_{y \in T(v)} \rho^T(x, y), \tag{7}$$

which is the optimal cost within the subtree. Suppose $v$ has more than one children $u, w, ...$, otherwise the optimal center is clear. Suppose the optimal solution of $cost_1^{T'}(v, U)$ chooses a leaf node in $T(u)$, and our HST initialization algorithm picks a leaf of $T(w)$. If $u = w$, then HST chooses the optimal one where the argument holds trivially. Thus, we consider $u \neq w$. We have the following two observations:

- Since one needs to pick a leaf of $T(u)$ to minimize $cost_1^{T'}(v, U)$, we have $cost_1^{T'}(v, U) \geq \sum_{x \in ch(v), x \neq u} N_x \cdot 2^{h_x}$ where $ch(u)$ denotes the children nodes of $u$.

- By our greedy strategy, $cost_1^T(v, U) \leq \sum_{x \in ch(u)} N_x \cdot 2^{h_x} \leq cost_1^{T'}(v, U) + N_u \cdot 2^{h_u}$.

Since $h_u = h_w$, we have

$$2^{h_u} \cdot (N_u - N_w) \leq 0,$$

since our algorithm picks subtree roots with highest scores. Then we have $cost_1^T(v, U) \leq cost_1^{T'}(v, U) + N_w \cdot 2^{h_w} \leq 2cost_1^{T'}(v, U)$. Since the subtrees in $C_1$ are disjoint, the union of centers for $OPT_1^T(v, U)$, $v \in C_1$ forms the optimal centers with size $k$. Note that, for any data point $p \in U \setminus C_1$, the tree distance $\rho^T(p, f)$ for $\forall f$ that is a leaf node of $T(v)$, $v \in C_1$ is the same. That is, the choice of leaf in $T(v)$ as the center does not affect the $k$-median cost under 2-HST metric. Therefore, union bound over $k$ subtree costs completes the proof. □

## C.3 PROOF OF PROPOSITION 3.2

*Proof.* It is known that the 2-HST can be constructed in $O(dn \log n)$ (Bartal, 1996). The subtree search in Algorithm 2 involves at most sorting all the nodes in the HST based on the score, which takes $O(n log n)$. We use a priority queue to store the nodes in $C_1$. When we insert a new node $v$ into queue, its parent node (if existing in the queue) would be removed from the queue. The number of nodes is $O(n)$ and each operation (insertion, deletion) in a priority queue based on score has $O(\log n)$ complexity. Lastly, the total time to obtain $C_0$ is $O(n)$, as the FIND-LEAF only requires a top down scan in $k$ disjoint subtrees of $T$. Summing parts together proves the claim. □

## C.4 PROOF OF THEOREM 4.2

Similarly, we prove the error in general metric by first analyzing the error in 2-HST metric. Then the result follows from Lemma 3.4. Let $cost_k^T(D)$, $cost_k^{T'}(D)$ and $OPT_k^T(D)$ be defined analogously to (3), (4) and (5), where "$y \in U$" in the summation is changed into "$y \in D$" since $D$ is the demand set. That is,

$$cost_k^T(D) = \sum_{y \in D} \min_{x \in C_0} \rho^T(x, y), \tag{8}$$

$$cost_k^{T'}(D, C_1) = \min_{|F \cap T(v)|=1, \forall v \in C_1} \sum_{y \in D} \min_{x \in F} \rho^T(x, y), \tag{9}$$

$$OPT_k^T(D) = \min_{F \subset D, |F|=k} \sum_{y \in D} \min_{x \in F} \rho^T(x, y) \equiv \min_{C_1'} cost_k^{T'}(D, C_1'). \tag{10}$$

We have the following.

**Lemma C.2.** $cost_k^T(D) \leq 10 OPT_k^T(D) + 10ck\epsilon^{-1}\triangle \log n$ *with probability* $1 - 4k/n^c$.

*Proof.* The result follows by combining the following Lemma C.4, Lemma C.5, and applying union bound. □

**Lemma C.3.** *For any node $v$ in T, with probability $1 - 1/n^c$, $|\hat{N}_v \cdot 2^{h_v} - N_v \cdot 2^{h_v}| \leq c\epsilon^{-1}\triangle \log n$.*

*Proof.* Since $\hat{N}_v = N_v + Lap(2^{(L-h_v)/2}/\epsilon)$, we have

$$Pr[|\hat{N}_v - N_v| \geq x/\epsilon] = exp(-x/2^{(L-h_v)}).$$

As $L = \log\triangle$, we have

$$Pr[|\hat{N}_v - N_v| \geq x\triangle/(2^{h_v}\epsilon)] \leq exp(-x).$$

Hence, for some constant $c > 0$,

$$Pr[|\hat{N}_v \cdot 2^{h_v} - N_v \cdot 2^{h_v}| \leq c\epsilon^{-1}\triangle \log n] \geq 1 - exp(-c\log n) = 1 - 1/n^c.$$

$\square$

**Lemma C.4** (DP Subtree Search). *With probability $1 - 2k/n^c$, $cost_k^{T'}(D) \leq 5OPT_k^T(D) + 4ck\epsilon^{-1}\triangle \log n$.*

*Proof.* The proof is similar to that of Lemma 3.5. Consider the intermediate output of Algorithm 2, $C_1 = \{v_1, v_2, ..., v_k\}$, which is the set of roots of the minimal disjoint subtrees each containing exactly one output center $C_0$. Suppose one of the optimal "root set" that minimizes (4) is $C_1^* = \{v_1', v_2', ..., v_k'\}$. Assume $C_1 \neq C_1^*$. By the same argument as the proof of Lemma 3.5, we consider for some $i, j$ such that $v_i \neq v_j'$, where $v_i$ is not a descendent of $v_j'$ and $v_j'$ is either a descendent of $v_i$. By the construction of $C_1$, we know that $score(v_j') \leq \min\{score(v_i), i = 1, ..., k\}$ when $v_j' \in C_1^* \setminus C_1$. Consider the swap between $C_1$ and $C_1^*$. By the definition of tree distance, we have $OPT_k^T(U) \geq \sum_{v_i \in C_1 \setminus C_1^*} N_{v_i} 2^{h_{v_i}}$, since $\{T(v_i), v_i \in C_1 \setminus C_1^*\}$ does not contain any center of the optimal solution determined by $C_1^*$ (which is also the optimal "root set" for $OPT_k^T$). Let us consider the optimal clustering with center set be $C^* = \{c_1^*, c_2^*, ..., c_k^*\}$ (each center $c_j^*$ is a leaf of subtree whose root be $c_j'$), and $S_j'$ be the leaves assigned to $c_j^*$. Let $S_j$ denote the set of leaves in $S_j'$ whose distance to $c_j^*$ is strictly smaller than its distance to any centers in $C_1$. Let $P_j$ denote the union of paths between leaves of $S_j$ to its closest center in $C_1$. Let $v_j''$ be the nodes in $P_j$ with highest level satisfying $T(v_j'') \cap C_1 = \emptyset$. The score of $v_j''$ is $2^{h_{v_j''}} N(v_j'')$. That means the swap with a center $v_j'$ into $C_1$ can only reduce $4 \cdot 2^{h_{v_j''}} N(v_j'')$ to $cost_k^{T'}(U)$ (the tree distance between any leaf in $S_j$ and its closest center in $C_1$ is at most $4 \cdot 2^{h_{v_j''}}$). We just use $v_j'$ to represent $v_j''$ for later part of this proof for simplicity. Summing all the swaps over $C_1^* \setminus C_1$, we obtain

$$cost_k^{T'}(U) - OPT_k^T(U) \leq 4 \sum_{v_j' \in C_1^* \setminus C_1} N_{v_j'} 2^{h_{v_j'}},$$

$$OPT_k^T(U) \geq \sum_{v_i \in C_1 \setminus C_1^*} N_{v_i} 2^{h_{v_i}}.$$

Applying union bound with Lemma C.3, with probability $1 - 2/n^c$, we have

$$N_{v_j'} 2^{h_{v_j'}} - N_{v_i} 2^{h_{v_i}} \leq 2c\epsilon^{-1}\triangle \log n.$$

Consequently, we have with probability, $1 - 2k/n^c$,

$$cost_k^{T'}(D) \leq 5OPT_k^T(D) + 4c|C_1 \setminus C_1^*|\epsilon^{-1}\triangle \log n$$
$$\leq 5OPT_k^T(D) + 4ck\epsilon^{-1}\triangle \log n.$$

$\square$

**Lemma C.5** (DP Leaf Search). *With probability $1 - 2k/n^c$, Algorithm 4 produces initial centers with $cost_k^T(D) \leq 2cost_k^{T'}(D) + 2ck\epsilon^{-1}\triangle \log n$.*

*Proof.* The proof strategy follows Lemma 3.6. We first consider one subtree with root $v$. Let $cost_1^{T'}(v, U)$ denote the optimal $k$-median cost within the point set $T(v)$ with one center in 2-HST:

$$cost_1^{T'}(v, D) = \min_{x \in T(v)} \sum_{y \in T(v) \cap D} \rho^T(x, y). \tag{11}$$

Suppose $v$ has more than one children $u, w, ...,$ and the optimal solution of $cost_1^{T'}(v, U)$ chooses a leaf node in $T(u)$, and our HST initialization algorithm picks a leaf of $T(w)$. If $u = w$, then HST chooses the optimal one where the argument holds trivially. Thus, we consider $u \neq w$. We have the following two observations:

- Since one needs to pick a leaf of $T(u)$ to minimize $cost_1^{T'}(v, U)$, we have $cost_1^{T'}(v, U) \geq \sum_{x \in ch(v), x \neq u} N_x \cdot 2^{h_x}$ where $ch(u)$ denotes the children nodes of $u$.

- By our greedy strategy, $cost_1^T(v, U) \leq \sum_{x \in ch(u)} N_x \cdot 2^{h_x} \leq cost_1^{T'}(v, U) + N_u \cdot 2^{h_u}$.

As $h_u = h_w$, leveraging Lemma C.3, with probability $1 - 2/n^c$,

$$2^{h_u} \cdot (N_u - N_w) \leq 2^{h_u}(\hat{N}_u - \hat{N}_w) + 2c\epsilon^{-1} \triangle \log n$$
$$\leq 2c\epsilon^{-1} \triangle \log n.$$

since our algorithm picks subtree roots with highest scores. Then we have $cost_1^T(v, D) \leq cost_k^{T'}(v, D) + N_w \cdot 2^{h_u} + 2c\epsilon^{-1} \triangle \log n \leq 2cost_k^{T'}(v, D) + 2c\epsilon^{-1} \triangle \log n$ with high probability. Lastly, applying union bound over the disjoint $k$ subtrees gives the desired result. $\qquad \square$

## C.5 PROOF OF THEOREM 4.3

*Proof.* The privacy analysis is straightforward, by using the composition theorem (Theorem C.1). Since the sensitivity of $cost(\cdot)$ is $\triangle$, in each swap iteration the privacy budget is $\epsilon/2(T + 1)$. Also, we spend another $\epsilon/2(T + 1)$ privacy for picking a output. Hence, the total privacy is $\epsilon/2$ for local search. Algorithm 4 takes $\epsilon/2$ DP budget for initialization, so the total privacy is $\epsilon$.

The analysis of the approximation error follows from Gupta et al. (2010), where the initial cost is reduced by our private HST method. We need the following two lemmas.

**Lemma C.6** (Gupta et al. (2010)). *Assume the solution to the optimal utility is unique. For any output $o \in O$ of $2\triangle\epsilon$-DP exponential mechanism on dataset $D$, it holds for $\forall t > 0$ that*

$$Pr[q(D, o) \leq \max_{o \in O} q(D, o) - (\ln|O| + t)/\epsilon] \leq e^{-t},$$

*where $|O|$ is the size of the output set.*

**Lemma C.7** (Arya et al. (2004)). *For any set $F \subseteq D$ with $|F| = k$, there exists some swap $(x, y)$ such that the local search method admits*

$$cost_k(F, D) - cost_k(F - \{x\} + \{y\}, D) \geq \frac{cost_k(F, D) - 5OPT(D)}{k}.$$

From Lemma C.7, we know that when $cost_k(F_i, D) > 6OPT(D)$, there exists a swap $(x, y)$ s.t.

$$cost_k(F_i - \{x\} + \{y\}, D) \leq (1 - \frac{1}{6k})cost_k(F_i, D).$$

At each iteration, there are at most $n^2$ possible outputs (i.e., possible swaps), i.e., $|O| = n^2$. Using Lemma C.6 with $t = 2 \log n$, for $\forall i$,

$$Pr[cost_k(F_{i+1}, D) \geq cost_k(F_{i+1}^*, D) + 4\frac{\log n}{\epsilon'}] \geq 1 - 1/n^2,$$

where $cost_k(F_{i+1}^*, D)$ is the minimum cost among iteration $1, 2, ..., t + 1$. Hence, we have that as long as $cost(F_i, D) > 6OPT(D) + \frac{24k \log n}{\epsilon'}$, the improvement in cost is at least by a factor of

$(1 - \frac{1}{6k})$. By Theorem 4.2, we have $cost_k(F_1, D) \leq C(\log n)(6OPT(D) + 6k\triangle \log n/\epsilon)$ for some constant $C > 0$. Let $T = 6Ck \log \log n$. We have that

$$E[cost(F_i, D)] \leq (6OPT(D) + 6k\epsilon^{-1}\triangle \log n)C(\log n)(1 - 1/6k)^{6Ck \log \log n}$$

$$\leq 6OPT(D) + 6k\epsilon^{-1}\triangle \log n \leq 6OPT(D) + \frac{24k \log n}{\epsilon'}.$$

Therefore, with probability at least $(1 - T/n^2)$, there exists an $i \leq T$ s.t. $cost(F_i, D) \leq 6OPT(D) + \frac{24k \log n}{\epsilon'}$. Then by using the Lemma C.7, one will pick an $F_j$ with additional additive error $4 \ln n/\epsilon'$ to the $\min\{cost(F_j, D), j = 1, 2, ..., T\}$ with probability $1 - 1/n^2$. Consequently, we know that the expected additive error is

$$24k\triangle \log n/\epsilon' + 4 \log n/\epsilon' = O(\epsilon^{-1}k^2\triangle(\log \log n) \log n),$$

with probability $1 - 1/poly(n)$.

$\square$

## D  EXTEND HST INITIALIZATION TO $k$-MEANS

Naturally, our HST method can also be applied to $k$-means clustering problem. In this section, we extend the HST to $k$-means and provide some brief analysis similar to $k$-median. We present the analysis in the non-private case, which can then be easily adapted to the private case. Define the following costs for $k$-means.

$$cost_{km}^T(U) = \sum_{y \in U} \min_{x \in C_0} \rho^T(x, y)^2, \tag{12}$$

$$cost_{km}^T{}'(U, C_1) = \min_{|F \cap T(v)| = 1, \forall v \in C_1} \sum_{y \in U} \min_{x \in F} \rho^T(x, y)^2, \tag{13}$$

$$OPT_{km}^T(U) = \min_{F \subset U, |F| = k} \sum_{y \in U} \min_{x \in F} \rho^T(x, y)^2 \equiv \min_{C_1'} cost_{km}^T{}'(U, C_1'). \tag{14}$$

For simplicity, we will use $cost_{km}^T{}'(U)$ to denote $cost_{km}^T{}'(U, C_1)$ if everything is clear from context. Here, $OPT_{km}^T$ (14) is the cost of the global optimal solution with 2-HST metric.

**Lemma D.1** (Subtree search). $cost_{km}^T{}'(U) \leq 17 OPT_{km}^T(U)$.

*Proof.* The analysis is similar with the proof of Lemma 3.5. Thus, we mainly highlight the difference. Let us just use some notations the same as in Lemma 3.5 here. Let us consider the clustering with center set be $C^* = \{c_1^*, c_2^*, ..., c_k^*\}$ (each center $c_j^*$ is a leaf of subtree whose root be $c_j'$), and $S_j'$ be the leaves assigned to $c_j^*$ in optimal k-means clustering in tree metric. Let $S_j$ denote the set of leaves in $S_j'$ whose distance to $c_j^*$ is strictly smaller than its distance to any centers in $C_1$. Let $P_j$ denote the union of paths between leaves of $S_j$ to its closest center in $C_1$. Let $v_j''$ be the nodes in $P_j$ with highest level satisfying $T(v_j'') \cap C_1 = \emptyset$. The score of $v_j''$ is $2^{h_{v_j''}} N(v_j'')$. That means the swap with a center $v_j'$ into $C_1$ can only reduce $(4 \cdot 2^{h_{v_j''}})^2 N(v_j'')$ to $cost_{km}^T{}'(U)$. We just use $v_j'$ to represent $v_j''$ for later part of this proof for simplicity. By our reasoning, summing all the swaps over $C_1^* \setminus C_1$ gives

$$cost_{km}^T{}'(U) - OPT_{km}^T(U) \leq \sum_{v_j' \in C_1^* \setminus C_1} N_{v_j'} \cdot (4 \cdot 2^{h_{v_j'}})^2,$$

$$OPT_{km}^T(U) \geq \sum_{v_i \in C_1 \setminus C_1^*} N_{v_i}(2^{h_{v_i}})^2.$$

Also, based on our discussion on Case 1, it holds that

$$N_{v_j'} 2^{h_{v_j'}} - N_{v_i} 2^{h_{v_i}} \leq 0.$$

Summing them together, we have $cost_{km}^T{}'(U) \leq 17 OPT_{km}^T(U)$.

$\square$

Next, we show that the greedy leaf search strategy (Algorithm 3) only leads to an extra multiplicative error of 2.

**Lemma D.2** (Leaf search). $cost_{km}^T(U) \leq 2cost_{km}^T{}'(U)$.

*Proof.* Since the subtrees in $C_1$ are disjoint, it suffices to consider one subtree with root $v$. With a little abuse of notation, let $cost_1^{T'}(v, U)$ denote the optimal $k$-means cost within the point set $T(v)$ with one center in 2-HST:

$$cost_1^{T'}(v, U) = \min_{x \in T(v)} \sum_{y \in T(v)} \rho^T(x, y)^2, \tag{15}$$

which is the optimal cost within the subtree. Suppose $v$ has more than one children $u, w, ...$, otherwise the optimal center is clear. Suppose the optimal solution of $cost_1^{T'}(v, U)$ chooses a leaf node in $T(u)$, and our HST initialization algorithm picks a leaf of $T(w)$. If $u = w$, then HST chooses the optimal one where the argument holds trivially. Thus, we consider $u \neq w$. We have the following two observations:

- Since one needs to pick a leaf of $T(u)$ to minimize $cost_1^{T'}(v, U)$, we have $cost_1^{T'}(v, U) \geq \sum_{x \in ch(v), x \neq u} N_x \cdot (2^{h_x})^2$ where $ch(u)$ denotes the children nodes of $u$.

- By our greedy strategy, $cost_1^T(v, U) \leq \sum_{x \in ch(u)} N_x \cdot (2^{h_x})^2 \leq cost_1^{T'}(v, U) + N_u \cdot (2^{h_u})^2$.

Since $h_u = h_w$, we have

$$2^{h_u} \cdot (N_u - N_w) \leq 0,$$

since our algorithm picks subtree roots with highest scores. Then we have $cost_1^T(v, U) \leq cost_1^{T'}(v, U) + N_w \cdot (2^{h_w})^2 \leq 2cost_1^{T'}(v, U)$. Since the subtrees in $C_1$ are disjoint, the union of centers for $OPT_1^T(v, U)$, $v \in C_1$ forms the optimal centers with size $k$. Note that, for any data point $p \in U \setminus C_1$, the tree distance $\rho^T(p, f)$ for $\forall f$ that is a leaf node of $T(v)$, $v \in C_1$ is the same. That is, the choice of leaf in $T(v)$ as the center does not affect the $k$-median cost under 2-HST metric. Therefore, union bound over $k$ subtree costs completes the proof. □

We are ready to state the error bound for our proposed HST initialization (Algorithm 2), which is a natural combination of Lemma D.1 and Lemma D.2.

**Theorem D.3** (HST initialization). $cost_{km}^T(U) \leq 34 OPT_{km}^T(U)$.

We have the following result based on Lemma 3.4.

**Theorem D.4.** *In a general metric space,*

$$E[cost_{km}(U)] = O(\min\{\log n, \log \triangle\})^2 OPT_{km}(U).$$

