# OpenReview forum: "k-Median Clustering via Metric Embedding: Towards Better Initialization with Differential Privacy"
_ICLR.cc/2023/Conference — Submitted to ICLR 2023_

### Official Review · Reviewer_2CeM · 2022-10-23

**Confidence:** 3
**Correctness:** 3
**Technical Novelty And Significance:** 2
**Empirical Novelty And Significance:** 2
**Recommendation:** 5

**Clarity, Quality, Novelty And Reproducibility:**

# Clarity:

The clarify is fine overall, but I have the following detailed comments.

1. Tree embedding in clustering-like problems have also been considered before, in e.g., “Facility location problems in differential privacy model revisited”, by Esencayi et al. Please add a comparison/discussion.
2. Also, I find “Differentially Private Clustering: Tight Approximation Ratios” by Ghazi et al. relevant but is not cited.
3. At the end of page one, you mentioned “k-median++”. It’s fine to call it “k-median++”, but it is actually “k-means++”, so one needs to clarify at least once, that k-median++ is the k-means++ adapted to the k-median case.
4. It seems you use \Delta as the diameter of the dataset. However, this makes sense only when you normalize the minimum distance between every pair of distinct point. Unfortunately, I didn’t see this mentioned (I might have missed it).
5. In the first bullet of page 2, you mentioned that \Delta = O(d) is the typical case of bounded data — I don’t agree, and in my opinion the bounded case should be \Delta = poly(n).
6. The definition of (2) is confusing, since the expression only replaces U with D. A suggestion is to avoid writing this (2) again, but simply say (1) with a differential privacy constraint is the DP k-median.
7. In the experiments, your refer to “left” and “right” column in Fig 2 and Fig 3. However, I only see one column. Maybe you can use sub-captions?

# Quality:

The major concern is the experiments.

1. The data is mostly from simulated sources, even though some of the simulation is based on real datasets. The suggestion is to experiment on more real datasets to make it more convincing. For instance, for the graph data, what about the road network data, such as OpenStreetMap? For Euclidean data, MNIST seems to be a small dataset, and experimenting a dataset of higher dimension and bigger size is helpful.

2. The running time comparison, which is an important aspect for initialization/seeding algorithms, is not provided.

# Originality:

The study of the general metric case is timely, but the techniques are somewhat standard, and I wouldn’t consider the result particularly novel since it is an immediate application of tree embedding, especially provided that similar ideas have been used in differential privacy (see e.g., “Facility location problems in differential privacy model revisited”, by Esencayi et al.).

**Strength And Weaknesses:**

# Strength:

While there are many recent works on the topic, I find the general metric case less studied, and this paper fills in this gap which is timely. The fact that the running time is independent of k can be crucial for some applications, and this is something k-means++ method cannot achieve. The tree embedding method is easy-to-implement, and is generally applicable, which is an advantage.

# Weakness:

The claimed improvement in ratio/error seems to be minor (for instance, log(\min{\Delta, k}) v.s. k-means++’s log k, and a log n -> log log n improvement in the number of iterations of local search, where there is already/still a factor of k^2).


**Summary Of The Paper:**

This paper studies initialization method that is based on tree embedding for k-median clustering, with differential privacy guarantee. By initialization, it means a light-weight approximation algorithm that finds k initial center points, to be used with iterative approximation algorithms such as local search. Since it is only an initialization, the approximation ratio does not need to be heavily optimized, and it is the efficiency that is important.

This paper focuses on the general metric space case. The main technical idea is to impose a tree embedding, which is O(log n) distortion to the true distance, and find a set of representative centers directly on the tree. The overall running time is \tilde{O}(nd), which is independent of k. The differential privacy can also be guaranteed, with a slight modification to the non-private version, by adding noise to some intermediate variables. The differentially private version, if combined with a previous work Gupta et al., can obtain a slightly improved additive error bound.

The experiments have been conducted to validate the performance of the new initialization method, including the widely-used k-means++. The new method demonstrates a better performance overall.


**Summary Of The Review:**

I would suggest a weak reject because of the limited technical novelty, and result-wise, the improvement over existing methods does not seem to be very significant.

---

> ### Author Response · Authors · 2022-11-19
> **Thank you for the review**
>
> Dear Reviewer 2CeM:
>
> 1. Thanks for pointing us to the reference. That paper also uses embedding trees in the algorithm design. However, the paper considers a different objective function than ours. Therefore, that paper and our paper are different in the way to add noise and update the centers. They did not use local search in the algorithm.
>
> 2. The paper considers continuous clustering problems. Here we consider discrete space. Hence, we believe that their result does not apply to our setting. We are happy to add this paper as reference.
>
> 3. Thank you. We will explain more on the naming and detail of k-median++.
>
> 4. We mentioned this in Definition 3.1.
>
> 5. We presented the running time comparison in Appendix B.2. One advantage of our method is that the running time does not depend on $k$. In applications where $k$ is large, our algorithm could be much more efficient.
>
> Thank you also for more suggestions to improve the quality of our paper.

---

### Official Review · Reviewer_SXQm · 2022-10-25

**Confidence:** 4
**Correctness:** 4
**Technical Novelty And Significance:** 2
**Empirical Novelty And Significance:** 1
**Recommendation:** 3

**Clarity, Quality, Novelty And Reproducibility:**

Most parts of the paper is well-written and the algorithm that solves the HST instance is novel.

The comparison with previous works is questionable in this paper. Most of the state of the art algorithms for k-median are not presented (for instance well-known publications based on LP roundings which result in the best approximation guarantee for k-median). Also more advanced algorithms are developed for solving an HST which is also not mentioned in this paper. Arthur and Vassilvitskii  paper in 2007 does not introduce k-median++ as well, it only focuses on k-means++.


**Strength And Weaknesses:**

non-private algorithm:
 - Strength:  The algorithm is well explained and the selection of centers in novel (to the best of my knowledge).
 - Weaknesses:
    - The approximation ratio of the algorithm is comparable with the HST based algorithm but significantly more than the best known algorithm.
    - There are almost linear time algorithms that find the optimum solution on a HST, the algorithm proposed in this work is significantly weaker (e.g., Parallel and Efficient Hierarchical k-Median Clustering). These works are not mentioned and compared at all in this paper.
    - The running time for metric space is not presented clearly, but it is presented for euclidean space.

Private algorithm:
 - Strength: the algorithm improves the additive error by a factor $O(\log n / \log \log n)$.
 - Weaknesses: the improvement in additive error is marginal given that there is more than a factor $k$ gap to the lower-bound.

Experiments:
 - The experiments consider only small size instances.
 - Experiments lack fast and state of the art k-median algorithms.
 - Experiments does ignore greedy k-median++ which is known to outperform k-median++.
 - HST is slower than k-median++ even for very small datasets, and seems to be worse as they grow.
 - There are few datasets considered.


**Summary Of The Paper:**

This work focuses on k-median clustering in metric space with privacy. The paper presents a new algorithm based on a HST and compares it with baselines in experiments.

1- The author presents a k-median clustering initialization with $O(\log \min{k,d})$ approximation guarantee.
2- They propose a differentially private algorithm with constant approximation guarantee and additive error $O(\frac{1}{\epsilon} \Delta k^2 \log^2 n)$.

Moreover the authors provide experiments for different datasets.


**Summary Of The Review:**

The algorithm is nice but it is not clear if it has any advantage in non-private settings and only a small factor improvement in the additive error in the differential private setting. This paper requires significant changes in the experimental section and comparison with previous work.

---

> ### Author Response · Authors · 2022-11-19
> **Thanks for your review**
>
> Dear Reviewer SXQm:
>
> Thanks for your comments on our paper.
>
> For general metric spaces, the best approximation ratio for k-Median is 2.611, as derived in
>
> [Jarosław Byrka, Thomas Pensyl, Bartosz Rybicki, Aravind Srinivasan, and Khoa Trinh. An
> improved approximation for k-median, and positive correlation in budgeted optimization, SODA 2014].
>
> We will cite this result in our paper. Thank you.
>
> Running time in metric space: In the paper, we presented Euclidean distance as an example. In metric space, k-median++ requires $O(km+kn\log n)$ time, while our method requires $O(m\log n+n\log n)$ time. We see that when $k>\log n$, k-median++ would be slower. This is the same as our claim in the Euclidean space.
>
> Regarding the experiments, we would like to clarify two points. Firstly, the main focus of our paper is to theoretically improve the approximation error of private k-median clustering in general metric space. Therefore, speed is not the major concern in our paper. We cited on page 4 a fast HST construction method as an example, and in the experiments we used the standard implementation for convenience. Second, in many practical problems, the number of clusters $k$ might be very large. We ourselves have worked on projects with millions of clusters (groups). The $O(k)$ vs $O(log(n))$ complexity of k-median++ and HST means that our algorithms would run much faster in this case. In our opinion, the independence in $k$ of the running time of our method is also one advantage.

---

### Official Review · Reviewer_RZVN · 2022-10-29

**Confidence:** 4
**Correctness:** 3
**Technical Novelty And Significance:** 3
**Empirical Novelty And Significance:** 3
**Recommendation:** 6

**Clarity, Quality, Novelty And Reproducibility:**

Clarity: Overall pretty clear.

But I encountered one irritating issue when reading the paper. The authors refer to $k$-medians++ as if it is a standard name for the technique they are using as the baseline. But AFAIK, this naming is very non-standard and I haven't seen such usage in the literature elsewhere. $k$-means++ gets its name because it is literally an augmented version of an algorithm which is referred to as the "$k$-means" algorithm. But the $k$-medians algorithm is not the local search algorithm, it refers to an algorithm similar to Lloyd's ($k$-means) but instead of computing the mean in the alternative minimization step, it computes the median of the group. See https://en.wikipedia.org/wiki/K-medians_clustering . Instead, I would refer to your baseline seeding algorithm as $D^1$-sampling following Wei (2016) : Wei, Dennis. "A constant-factor bi-criteria approximation guarantee for k-means++." Advances in Neural Information Processing Systems 29 (2016).

A Constant-Factor Bi-Criteria Approximation Guarantee for k-means++

Another issue was that it was a bit hard to compare the plots. They were too crowded. I suggest providing plots for the non-DP algorithms and the DP-algorithms separately in the appendix.

Quality: The submission is technically sound. All claims are well-supported with proofs and detailed experiments.

Novelty might be the Achilles' heel for this paper.

Reproducibility: Proofs are provided. The code is not provided but their algorithms and experiments are detailed enough that it would not be too hard to reproduce their results. But I encourage the authors to provide the code as well (or open-source it).

----------------------------------------------
Typos/minor issues:

Line just before the inequality in the introduction: “a clustering algorithms”

In section 2.1, you claim that Arya et al. (2004) showed that cost(F) $\leq$ 5OPT for the Algorithm 1 you describe. But that is only true when $\alpha = 0$ i.e. if the final solution is a locally-optimal solution. There should be a term dependent on alpha. Something like cost(F) $\leq 5(1+\alpha)$ OPT but I am not entirely sure about the constant (might be $5(1+2\alpha)$ or something like that). I wouldn't bother too much with this. You can just provide a note. If you are enthusiastic enough and want to compute the exact constant, I would Williamson-Shmoys chapter 9 over Arya et al.

Experiments 5.1 “Discrete Euclidean space. Following previous work .,” has .,

**Strength And Weaknesses:**

Strength: The paper is overall nicely written and studies important problems.

Weaknesses: Although their algorithm is interesting academically, I don't find either their approximation bounds or their experimental results to be earth shattering.

**Summary Of The Paper:**

The paper provides a seeding technique for the local search algorithm for k-median clustering in general metric spaces. Their algorithm is based on a tree embedding of the data. They also provide a version of their algorithm which can be used for differentially private $k$-median clustering.

**Summary Of The Review:**

Clean and easy paper to accept, but nothing groundbreaking.

---

> ### Author Response · Authors · 2022-11-19
> **Thanks for your review**
>
> Dear Reviewer RZVN:
>
> Thank you for your positive feedback and constructive suggestions. We will consider modifying the name of "k-median++" and discuss more details about it (e.g., the relationship to k-means++). We have corrected the typos. We will also try to modify the plots. Thanks also for suggesting the book by Williamson and Shmoys.
>
> Again, we sincerely appreciate your valuable feedback.

---

### Official Review · Reviewer_it5w · 2022-11-04

**Confidence:** 3
**Clarity, Quality, Novelty And Reproducibility:** The paper is written clearly.
**Correctness:** 4
**Technical Novelty And Significance:** 2
**Empirical Novelty And Significance:** 2
**Recommendation:** 3

**Strength And Weaknesses:**

The claims are somewhat dubious, because the paper introduces a new parameter (the diameter of the metric space). The bound becomes better if this parameter is small, but it is not clear that previous analyses of kmeans++ variants cannot be proven to do better if this parameter is taken into account. Moreover, part of the attractiveness of the older seeding schemes does not lie in their worst case performance, but rather in their excellent performance in stable instances, under various notions of stability.

The paper has some empirical evidence that their method is good. It's beyond my expertise to judge the empirical part.

Also, I cannot evaluate the significance of the differential privacy result.

The survey of relevant literature is lacking.

**Summary Of The Paper:**

The paper proposes to seed Lloyd's iteration for computing k-means clustering, in the context of finite metric spaces, using the approximation the underlying metric by a convex combination of hierarchically separated tree metrics. This is claimed to achieve better approximation guarantees (for the seeding step) than current conventions.

Also, the same idea is used to generate differentially private seeding.

**Summary Of The Review:**

The paper needs to address the comparison with previous methods more seriously.

---

> ### Author Response · Authors · 2022-11-19
> **Thanks for your review.**
>
> Dear Reviewer it5w:
>
> Thanks for your feedback on our paper. Our problem setup is consistent with the original paper of [Gupta et al., 2010] to consider the worst case bounds in both non-private and private setting. If we add stability assumption to our problem, we expect that our error rate can also be improved. This might be a good future direction.
>
> The empirical results and the design and analysis of DP-HST are indeed both important contributions of our paper. On this line of research (private clustering in metric space), we are not aware of any previous work that provided numerical results. As such setup can be common in practice (e.g., traffic graphs, maps, social network, etc.), we feel that our empirical results are also a good contribution to the community on the practicality of related methods.
>
> Again, we thank you for the valuable comments.

---

### Author Response · Authors · 2022-12-10
**Follow up**

Dear Reviewers and Area Chair:

Once again, we appreciate your valuable feedback on our work.

One of the main concerns from the reviews is the comparison with some more methods in the literature. Our paper is the first one which rigorously improves the metric DP local search algorithm in [Gupta et al., 2010]. Therefore, our results are not comparable with many works considering clustering in the continuous Euclidean space. Additionally, in this line of research, we are not aware of any previous works that also provided empirical study. Hence, we have hoped that our empirical results (which did not exist in the literature) in the submission could also contribute to the understanding of DP local search algorithms in practice.

Please let us know if Reviewers might have additional questions.

---

### Decision · Program_Chairs · 2023-01-20

**Decision:**

Reject

**Justification For Why Not Higher Score:**

The contributions do not seem to be sufficiently significant or novel.

**Justification For Why Not Lower Score:**

N/A

**Metareview: Summary, Strengths And Weaknesses:**

The paper introduces a new method for constructing an initialization for k-median clustering in general metric spaces. The paper embeds the metric space into a hierarchically-separated tree and it uses sampling from the HST in order to select the initial cluster centers. The initialization approach can be readily used to obtain differentially private clusterings. The theoretical analysis of the approximation guarantee gives a guarantee in terms of the diameter of the metric space, and this bound improves upon existing guarantees if the diameter is sufficiently small. In the private setting, the paper achieves an approximation guarantee with an improved additive error.

The reviewers were concerned about the novelty and strength of the theoretical contribution. In the non-private setting, the improvement in the approximation guarantee only holds when the diameter is small, and it is not clear whether this is a true improvement over prior work since it is possible that the prior analyses could also give an analogous improvement in the small diameter regime. In the private setting, there is only a small improvement in the additive error. Overall, there was consensus among the reviewers that the contributions are not sufficiently significant or novel to warrant acceptance.